# Single and Combined Effects of Cannabigerol (CBG) and Cannabidiol (CBD) in Mouse Models of Oxaliplatin-Associated Mechanical Sensitivity, Opioid Antinociception, and Naloxone-Precipitated Opioid Withdrawal

**DOI:** 10.3390/biomedicines12061145

**Published:** 2024-05-22

**Authors:** Sean A. Hayduk, Amanda C. Hughes, Rachel L. Winter, Mia D. Milton, Sara Jane Ward

**Affiliations:** Center for Substance Abuse Research, Department of Neural Sciences, Lewis Katz School of Medicine, Temple University, Philadelphia, PA 19140, USA; sean.hayduk@temple.edu (S.A.H.); amanda.hughes@temple.edu (A.C.H.); rachel.winter@temple.edu (R.L.W.); mia.milton@temple.edu (M.D.M.)

**Keywords:** cannabinoid, cannabidiol, cannabigerol, CIPN, pain, oxaliplatin, opioid withdrawal

## Abstract

Chemotherapy-induced peripheral neuropathy (CIPN) is one of the most prevalent and dose-limiting complications in chemotherapy patients, with estimates of at least 30% of patients experiencing persistent neuropathy for months or years after treatment cessation. An emerging potential intervention for the treatment of CIPN is cannabinoid-based pharmacotherapies. We have previously demonstrated that treatment with the psychoactive CB1/CB2 cannabinoid receptor agonist Δ^9^-tetrahydrocannabinol (Δ^9^-THC) or the non-psychoactive, minor phytocannabinoid cannabidiol (CBD) can attenuate paclitaxel-induced mechanical sensitivity in a mouse model of CIPN. We then showed that the two compounds acted synergically when co-administered in the model, giving credence to the so-called entourage effect. We and others have also demonstrated that CBD can attenuate several opioid-associated behaviors. Most recently, it was reported that another minor cannabinoid, cannabigerol (CBG), attenuated cisplatin-associated mechanical sensitivity in mice. Therefore, the goals of the present set of experiments were to determine the single and combined effects of cannabigerol (CBG) and cannabidiol (CBD) in oxaliplatin-associated mechanical sensitivity, naloxone-precipitated morphine withdrawal, and acute morphine antinociception in male C57BL/6 mice. Results demonstrated that CBG reversed oxaliplatin-associated mechanical sensitivity only under select dosing conditions, and interactive effects with CBD were sub-additive or synergistic depending upon dosing conditions too. Pretreatment with a selective α2-adrenergic, CB1, or CB2 receptor selective antagonist significantly attenuated the effect of CBG. CBG and CBD decreased naloxone-precipitated jumping behavior alone and acted synergistically in combination, while CBG attenuated the acute antinociceptive effects of morphine and CBD. Taken together, CBG may have therapeutic effects like CBD as demonstrated in rodent models, and its interactive effects with opioids or other phytocannabinoids should continue to be characterized.

## 1. Introduction

As the opioid crisis continues in the US, those suffering from chronic pain look to safer and more effective treatment options, including natural remedies. An example of a difficult to treat chronic pain condition is chemotherapy-induced peripheral neuropathy (CIPN). To date, no drug or drug class is safe and effective for treatment of CIPN-associated pain [1], making the identification of alternative effective analgesics a crucial medical need. CIPN is one of the most prevalent and dose-limiting complications in chemotherapy patients. The incidence of CIPN specifically associated with oxaliplatin therapy can approach 75% with certain regimens, and toxicity can be so severe as to lead to cessation of cancer treatment. Common peripheral sensory symptoms include paresthesias and dysesthesias, pain, numbness and tingling, and sensitivity to touch and temperature. Damage to peripheral nerves can lead to spontaneous sensory neuron activity [2], alteration of voltage-gated ion channel activity [3,4], ascending fiber pathology [5], and neuroinflammation [6], ultimately leading to ascending pain pathway sensitization [7]. Functional changes to the descending inhibitory pain pathway can also result, altering serotonin and noradrenergic signaling and further amplifying the effects of central sensitization [8,9].

The scientific evidence, practice, and legislation surrounding the medical use of Cannabis for the treatment of chronic pain continues to rise, and there is growing patient and research interest in cannabinoid constituents for the treatment of CIPN (see [10] for review). We and others have demonstrated that both Δ^9^-tetrahydrocannabinol (Δ^9^-THC) and cannabidiol (CBD) can prevent the development of mechanical sensitivity associated with chemotherapeutic administration in male [11,12] or female [13,14] C57BL/6 mice. We demonstrated that the effect of CBD likely involved activation of the serotonin 5-HT1A receptor, as antagonism of this receptor (but not the cannabinoid CB1 or CB2 receptor) negated the protective effect of CBD [14]. Subsequently, we established that CBD works synergistically with Δ^9^-THC against CIPN, increasing the potency to attenuate mechanical sensitivity ten-fold [11]. More recently, it was demonstrated that the phytocannabinoid cannabigerol (CBG) significantly reduced mechanical hypersensitivity in a mouse model of cisplatin-associated peripheral neuropathy [15,16]. Limited binding studies have addressed potential mechanisms of action of CBG. It has been described as an antagonist at 5-HT1A receptors and an agonist at the α2 adrenoreceptor [17], and as a weak partial agonist of CB1R and CB2R [18]. Sepulveda et al. [15] reported that the antinociceptive effects of CBG in the mouse model of cisplatin-induced peripheral neuropathy were attenuated by either α2 -adrenergic, CB1, or CB2 receptor antagonism. Taken together, these results strongly suggest that CBD and CBG share partially overlapping receptor interactions, and further work is needed to understand their anti-neuropathic effects alone and in combination.

Another line of research has focused on whether combining cannabinoids with opioid dosing regimens will make opioid treatment safer and/or more effective. For example, it has been established in several rodent models that CB1 receptor agonists such as Δ^9^-THC can increase the potency of opioid antinociceptive effects [19,20,21,22], but see [23] or attenuate the development of opioid tolerance [24]; (but not in monkeys, see [25]). Less is known about the effects of the so-called minor cannabinoids on opioid antinociception, tolerance, and dependence. We reported in Neelakantan et al. [26] that CBD enhanced the antinociceptive effect of morphine in the acetic acid stretching test but attenuated morphine-associated thermal antinociception as measured by the hotplate, demonstrating robust endpoint specific interactive effects. Others have shown that CBD can attenuate opioid withdrawal signs in rodents [27,28]. We are unaware of any published work investigating CBG/opioid interactions.

Therefore, the goals of the present set of experiments were to determine the single and combined effects of CBG and CBD in reversal of oxaliplatin-associated mechanical sensitivity, attenuation of naloxone-precipitated morphine withdrawal, and acute morphine antinociception in male C57BL/6 mice. Male C57BL/6 mice were used for the present study. A separate study focusing on investigating potential sex differences in the behavioral and neuroinflammatory responses to CBG in CIPN models has also been conducted by our laboratory and presented elsewhere [29].

## 2. Materials and Methods

### 2.1. Animals

The animal experiments presented in this article were approved by the Institutional Animal Care and Use Committee (IACUC; ACUP4914) at Temple University (Philadelphia, PA, USA). A total of 640 mice were used in this study. All experiments began with 8 mice per group. No animals were excluded from the data analysis. In some experimental groups, mice did not complete the experiment due to fighting wounds resulting from group housing, which can occur with male mice. The mice used in these experiments were purchased from Taconic Biosciences (Germantown, NY, USA). They were male C57BL/6NTac mice, aged 6–8 weeks at time of arrival to the vivarium. All mice acclimated to the vivarium for at least 5 days prior to initiation of behavioral testing. Mice were maintained in an enriched environment with a dark/light cycle of 12 h and a temperature of 22 °C. Mice were housed 4 per cage and had ad libitum access to regular food and water. The behavioral experiments on the mice were performed during the light cycle. Every effort was made to ensure optimal welfare conditions before, during, and after each experiment, and the mice were observed daily for general condition. The size of the animal groups for the experiments was based on data from previous studies. The observer of the behavioral tests was not aware of the treatment of the animals. Mice were randomly assigned to their groups.

### 2.2. Drugs

Oxaliplatin (Pfizer Hospital) was procured from Temple University Pharmacy. Oxaliplatin was dissolved in a mixture of Kolliphor (Sigma-Aldrich, St. Louis, MO, USA), ethanol (Sigma-Aldrich, St. Louis, MO, USA), and saline (KD Medical, Columbia, MD, USA) (mixture proportion 1:1:18). Intraperitoneal injections of oxaliplatin were performed once at a dose of 6.0 mg/kg. Control mice received the vehicle (1:1:18, ethanol, Kolliphor, and saline) at a volume of 10 mL/kg, i.p. CBG was generously provided by Benuvia Pharmaceuticals, Round Rock, TX, USA. CBG was dissolved in a mixture of Kolliphor (Sigma-Aldrich, St. Louis, MO, USA), ethanol (Sigma-Aldrich, St. Louis, MO, USA), and saline (mixture proportion 1:1:18) and was administered in a range of dosing regimens described below. CBD was purchased from Cayman Chemical (Ann Arbor, MI, USA). CBD was dissolved in a mixture of Kolliphor (Sigma-Aldrich, St. Louis, MO, USA), ethanol (Sigma-Aldrich), and saline (mixture proportion 1:1:18) and was administered in a range of dosing regimens described below. Morphine was obtained through the NIDA Drug Supply Program and was dissolved in 0.9% sodium chloride (Hospira, Lake Forest, IL, USA). Naloxone hydrochloride was purchased from Enzo Life Sciences (Farmingdale, NY, USA). Atipamezole was dissolved in 0.9% sodium chloride (Hospira). SR141716 and SR144528 were dissolved in a mixture of Kolliphor (Sigma-Aldrich, St. Louis, MO, USA), ethanol (Sigma-Aldrich), and saline (mixture proportion 1:1:18). All three antagonists were purchased from (Sigma-Aldrich, St. Louis, MO, USA). All doses were selected based on our published work with cannabinoids and opioids in these models [11,12,13,14], as well as the previous work with CBG in CIPN models in the literature [15,16].

### 2.3. Mechanical Sensitivity

#### 2.3.1. von Frey Filaments Test

Baseline mechanical sensitivity testing took place for three consecutive days (Days −2, −1, and 0) before administration of oxaliplatin on Day 1. During baseline testing and then again on Day 6 (test day), mice were placed in individual Plexiglas compartments (Med Associates, St. Albans, VT, USA) on top of a wire grid floor suspended 20 cm above the laboratory bench top and acclimatized to the environment for 30 min. Mechanical allodynia was assessed using von Frey monofilaments of varying forces (0.07–2.0 g) applied to the mid-plantar surface of the right hind paw, with each application held in c-shape for 6 s, starting with the 0.07 filament. If no response was elicited, the next filament was tested until a response was elicited. Filaments were then retested in a descending order until the filament did not elicit a response, and the lowest filament to elicit a response was recorded [11,12,13,14]. CBG, CBG, or their combination was administered in a series of dosing regimens, from only on Day 6, 2 h prior to the test session, to Days 5 and 6, Days 4–6, Days 3–6, and Days 2–6. For antagonist studies, the antagonist was administered 30 min prior to CBG.

#### 2.3.2. Naloxone-Precipitated Jumping Behavior

Morphine dependence was induced by twice daily injections (9 a.m. and 7 p.m.) for five consecutive days using an escalating dose schedule (twice daily 20 mg/kg, 40 mg/kg, 60 mg/kg, 100 mg/kg). On Day 5, mice received a final dose of morphine (100 mg/kg i.p.) and injection of vehicle, CBD, or CBG (10–100 mg/kg) and placed back into their home cages to allow for habituation. After 2 h, all mice were challenged with naloxone (3 mg/kg i.p.) and were immediately placed into individual Plexiglas observation chambers to measure precipitate a μ-opioid receptor-dependent withdrawal syndrome, as described previously [30]. Mice were observed for 30 min and the number of withdrawal jumps following the naloxone challenge was quantified by a blinded scorer.

### 2.4. Hotplate Antinociception

The effects of CBD, morphine, and CBG alone and CBG + morphine or CBG + CBD were assessed using a hot plate thermal nociceptive assay. In this procedure, a single mouse was placed on the hot plate set to a temperature of 56 °C with a 15 cm high plastic cylinder. The latency to when the mouse licked its hind paw, jumped off the hot plate, or a cutoff time of 20 s was recorded. Three baseline latencies spaced 5 min apart were determined and the mean value for each individual mouse was taken as the baseline latency measure. Mice were administered CBG, CBD, or morphine (1.0–30 mg/kg) and tested on the hot plate 30 min later to determine post-drug latencies. Combinations of CBG with morphine or CBD by retesting the morphine or CBD dose response but keeping the CBG dose set to 30 mg/kg. Each mouse served as its own control in the experiments. For the data analyses, hot plate latencies after drug administration were expressed as a % maximal possible effect (%MPE) and calculated by the following equation: %MPE = [Test (post-drug) latency − baseline latency]/[Cutoff (20 s) − baseline latency] × 100. A value of 0 was assigned if the mouse responded faster after drug administration than its average baseline latency. The %MPE was calculated for each dose of the drugs tested alone or in combination for individual mice and then averaged into a group mean.

### 2.5. Statistical Analyses

One-way ANOVA, curve fitting, and linear regressions for statistical analyses were performed on dose–response data using GraphPad Prism 10 (GraphPad Prism 10.0 Software Inc., La Jolla, CA, USA). In all three behavioral assays, the dose–response effects for CBG, CBD, and morphine (where applicable) were analyzed using one-way ANOVA with Dunnett’s multiple comparisons post hoc tests.

### 2.6. Combination Analyses

Expected effects of drug combinations were calculated for all dose combinations in the three behavioral tests based on the principles of dose addition. As described by Tallarida (2000) [31], regression equations were calculated for each set of individual dose–response curves using linear regression analysis to determine equipotent dose ratios of CBG, CBD, and morphine for the behavioral assays. The choice of doses for the combinations from the linear relation is typically made by starting with the individual ED_50_ values when maximal effects are similar, and the potency of each agent is relatively constant over the effect range. The expected ED_50_ value is then determined to be the effect of combining the ED_25_ of each compound alone when they are tested in combination. This equipotent dose ratio, as well as lower and higher dose combinations set at the same ratio to one another, are then tested to determine the observed ED_50_ value. In the case where the single compounds are not efficacious enough to determine ED_50_ values, other effect levels can be determined and used; therefore, for naloxone-precipitated jumping behavior, ED_25_ values were calculated and used. Lastly, in cases in which the maximum effects of two compounds differ, as in the case of morphine or CBD versus CBG in the hotplate assay, the dose choice paired the ED_50_ of the higher efficacy drug with the dose of the other that produces a maximal effect.

If the observed effect was significantly greater than the expected effect, the drug combination in the studied dose ratio would be considered synergistic. If the observed effect was significantly less than the expected effect, the drug combination in the studied dose ratio would be considered sub-additive. No significant differences between the expected and observed effects would be considered simply as an additive interaction.

## 3. Results

In the first experiment, vehicle or oxaliplatin was administered on Day 1, and CBG or CBD was administered on Day 6, followed by mechanical sensitivity measurement. We observed that administration of CBD, but not CBG, on Day 6 reversed mechanical sensitivity following oxaliplatin exposure (Figure 1A,B). For CBG, one-way ANOVA revealed a significant effect of treatment [F_(4,35)_ = 11.06], *p* < 0.0001, and Dunnett’s multiple comparisons test revealed that only the vehicle treatment group was statistically different from the oxaliplatin alone treatment, showing that CBG administration did not attenuate mechanical sensitivity associated with oxaliplatin exposure. For CBD, one-way ANOVA revealed a significant effect of treatment [F_(4,35)_ = 3.537], *p* < 0.01, and Dunnett’s multiple comparisons test revealed that the 30 mg/kg CBD treatment group was statistically different from the oxaliplatin alone treatment, showing that CBD administration attenuated mechanical sensitivity associated with oxaliplatin exposure in a dose-dependent manner.

To determine equipotent dose ratios of CBG and CBD to test in combination and to determine the expected additive effect level of the combination, data were transformed to percent baseline mechanical sensitivity and ED_50_ values were calculated for CBG and CBD. The ED_50_ value for CBG was determined to be 40.67 mg/kg, and the ED_50_ value for CBD was determined to be 19.1 mg/kg (Figure 1C,D). To test the combination, mice were treated with vehicle or oxaliplatin on Day 1 and increasing dose combinations of CBG and CBD in a 1:1 ratio based on potency determined by the single compound ED_50_ values, and the observed ED_50_ value was determined to be 3.58 mg/kg CBG + 1.68 mg/kg CBD, or 5.26 mg/kg (Figure 1E), compared to the expected ED_50_ value of 20.34 mg/kg CBG + 9.55 mg/kg CBD, or 29.89 mg.kg. The singe compound ED_50_ values can be plotted on an isobologram with a predicted line of additivity connecting them; observed ED_50_ values that lay along the line of additivity demonstrate additive combination effects. The observed ED_50_ value of 5.26 mg/kg lays well within the zone of synergy as shown on the isobologram (Figure 1F).

In the next experiment, increasing days of CBG injections were tested to determine whether repeated CBG administration would produce an effect on oxaliplatin-associated mechanical sensitivity. The only dosing regimen that produced a significant effect on mechanical sensitivity was CBG injections on Days 4, 5, and 6 post-oxaliplatin exposure (Figure 2). One-way ANOVA showed a significant effect of treatment [F_(4,35)_ = 6.599], *p* = 0.005, with Dunnett’s post-test comparison showing a significant difference between the 10 mg/kg CBG treated group and the oxaliplatin alone group. Treatment on Days 5 and 6 [F_(4,35)_ = 7.267], 3, 4, 5, and 6 [F_(4,35)_ = 6.303], or 2, 3, 4, 5, and 6 [F_(4,35)_ = 11.06] only showed a significant effect of oxaliplatin alone versus vehicle, with no post hoc effects of CBG treatment at any dose.

We then sought to determine the interactive effect of CBG and CBD at this dosing regimen that produced an effect for CBG. We first determined that dosing of CBD on days 4, 5, and 6 was effective at reversing oxaliplatin-associated mechanical sensitivity comparable to CBG (Figure 3A,B). For CBD, one-way ANOVA revealed a significant effect of treatment [F_(4,35)_ = 7.979], *p* = 0.0001, and Dunnett’s multiple comparisons test revealed that the 30 mg/kg CBD treatment group was statistically different from oxaliplatin alone treatment, showing that CBD administration attenuated mechanical sensitivity associated with oxaliplatin exposure in a dose-dependent manner.

For the Days 4, 5, and 6 dosing regimens, the ED_50_ value for CBG was determined to be 8.0 mg/kg, and the ED_50_ value for CBD was determined to be 4.4 mg/kg (Figure 3C,D). To test the combination, mice were treated with vehicle or oxaliplatin on Day 1 and increasing dose combinations of CBG and CBD in a 1:1 ratio based on potency determined by the single compound ED_50_ values on Days 4, 5, and 6, and the observed ED_50_ value was determined to be 14.8 mg/kg CBG + 8.17 mg/kg CBD, or 22.97 mg/kg (Figure 3E), compared to the expected ED_50_ value of 4.0 mg/kg CBG + 2.2 mg/kg CBD, or 6.2 mg/kg. The singe compound ED_50_ values were plotted on the isobologram with a predicted line of additivity connecting them; the observed ED_50_ value of 22.97 mg/kg lays well within the zone of subadditivity as shown on the isobologram (Figure 3F).

We next determined the effect of selective antagonists for α2-adrenergic (atipamezole), CB1 (SR141716), or CB2 (SR144528) receptors on CBG-attenuation of oxaliplatin mechanical sensitivity using the Days 4, 5, and 6 dosing regimens. All three antagonists dose dependently attenuated CBG’s effect (Figure 4). For atipamezole, one-way ANOVA revealed a significant effect of treatment [F_(5,40)_ = 6.356], *p* = 0.0002, with Dunnett’s multiple comparison test showing a significant effect of oxaliplatin alone, atipamezole 3.0 mg/kg, and atipamezole 10 mg/kg as compared with vehicle/vehicle treated. For SR141716, one-way ANOVA revealed a significant effect of treatment F_(5,42)_ = 5.615], *p* = 0.0005, with Dunnett’s multiple comparison test showing a significant effect of oxaliplatin alone, SR141716 0.3 mg/kg, SR141716 1.0 mg/kg, and SR141716 3.0 mg/kg. For SR144528, one-way ANOVA revealed a significant effect of treatment F_(5,42)_ = 4.587], *p* = 0.0020, with Dunnett’s multiple comparison test showing a significant effect of oxaliplatin alone, SR144528 1.0 mg/kg, and SR144528 3.0 mg/kg.

We next determined the single and combined effects of CBG or CBD on naloxone-precipitated withdrawal in morphine dependent mice. Results demonstrate that 100 mg/kg CBG or CBD administration significantly attenuated naloxone-precipitated jumping (Figure 5A,B). For CBG, one-way ANOVA showed an effect of treatment [F_(3,35)_ = 3.403], *p* = 0.0282, with Dunnett’s multiple comparison test showing a significant effect of 100 mg/kg CBG. For CBD, one-way ANOVA showed an effect of treatment [F_(3,32)_ = 4.179, *p* = 0.0133, with Dunnett’s multiple comparison test showing a significant effect of 100 mg/kg CBD. For attenuation of naloxone-precipitated jumping, the ED_25_ value for CBG was determined to be 21.7 mg/kg, and the ED_25_ value for CBD was determined to be 26.4 mg/kg (Figure 5C,D). To test the combination, mice were treated with morphine for the last injection, along with vehicle or increasing dose combinations of CBG and CBD in a 1:1 ratio based on potency determined by the single compound ED_25_ values. The observed ED_25_ value was determined to be 2.48 mg/kg CBG + 3.02 mg/kg CBD, or 5.5 mg/kg (Figure 5E), compared to the expected ED_25_ value of 10.85 mg/kg CBG + 13.2 mg/kg CBD, or 24.05 mg/kg. The singe compound ED_25_ values were plotted on the isobologram with a predicted line of additivity connecting them; the observed ED_25_ value of 5.5 mg/kg lays well within the zone of synergy as shown on the isobologram (Figure 5F). 

In the final set of experiments, we determined the single and combined effects of CBG, CBD, and morphine on thermal antinociception using the hotplate (Figure 6). Results demonstrate that CBG and CBD produce mild to moderate antinociception on the hotplate, respectively, while morphine produces robust dose-dependent antinociceptive effects, as expected. We next tested whether the 30 mg/kg dose of CBG would modify the antinociceptive effects of morphine or CBD. For morphine, two-way ANOVA revealed that addition of CBG produced a significant attenuation of morphine antinociception. There was a significant main effect of drug [F_(1,65_) = 5.236], *p* = 0.0254 and of dose [F_(4,65)_ = 59.39], *p* < 0.0001, but no significant interaction [F_(4,65)_ = 1.686]. Sidak’s multiple comparisons test revealed a significant difference between morphine versus morphine + CBG at the 30 mg/kg dose. For CBD, two-way ANOVA revealed that addition of CBG did not alter the modestly observed CBD antinociception. There was a significant main effect of dose [F_(4,65)_ = 5.629], *p* = 0.0006, but not drug [F_(1,65)_ < 1.0], and no significant interaction [F_(4,65)_ = 1.556]. However, there was a non-significant attenuation of CBD by CBG at the 100 mg/kg CBD dose.

## 4. Discussion

In the present set of experiments, we replicated our previous observations that CBD administration attenuates oxaliplatin-associated mechanical sensitivity in male C57BL/6 mice [11] and extended these findings to show that that effect is observed under single as well as repeated administration conditions (Figure 1 and Figure 3). In contrast, we found that administration of CBG was able to attenuate oxaliplatin-associated mechanical sensitivity under a much narrower set of dose and dosing conditions (Days 4–6 post oxaliplatin exposure; Figure 1 and Figure 2). Previously, both acute and chronic administration of CBG were shown to reverse an established mechanical sensitivity to cisplatin in male and female mice [15,16] following single, 7, or 14 days of injection. As mentioned previously, CBG has been described as an antagonist at 5-HT1A receptors and an agonist at the α2 adrenoreceptor [17], and as a weak partial agonist of CB1R and CB2R [18]. Under the acute dosing condition, Sepulveda et al. [15] showed that the effect of CBG could be attenuated by antagonism of the CB1 or CB2 receptor, as well as the α2-adrenergic receptor, and we replicate this observation under the Days 4, 5, and 6 dosing regimens. These data taken together could suggest that CBG may be less effective at attenuating mechanical sensitivity associated with oxaliplatin than with cisplatin, although the mechanisms may be similar. The other methodologies between the studies from our laboratory and that of Sepulveda [15] and Nachnani [16] are very similar, but it is possible that the disparate effects were due to other factors than the chemotherapeutic agent used.

Our previous work also demonstrated that CBD works synergistically with Δ^9^-THC against paclitaxel associated mechanical threshold, increasing the potency to attenuate touch sensitivity ten-fold [11]. In the present study, we tested the nature of interaction between CBD and CBG under two dosing conditions: one where both CBD and CBG were effective, and one where only CBD was effective. Under the Days 4, 5, 6 dosing regimens, where both phytocannabinoids alone attenuated mechanical sensitivity, the combination produced subadditive effects (Figure 4). This addition of CBD to CBG leading to a reduced potency could be seen as similar to the effect that was observed with CBG alone, where higher doses of CBG (30 mg/kg) or more days of CBG administration, equated to a loss of therapeutic effect. One mechanistic explanation that comes to mind from this is the role of the 5-HT1A receptor in pain modulation. We showed previously that the protective effect of CBD in the paclitaxel mechanical sensitivity assay was mediated by the 5HT1A receptor [14], in that blockade of this receptor prevented the therapeutic effect we observed with CBD. In contrast, limited evidence has suggested that CBG itself is a putative 5-HT1A receptor antagonist [17]. Therefore, this one component of CBG’s pharmacological effect is counter to our conceptualization of the role of 5-HT1A and pain inhibition and may contradict its more therapeutic mechanisms of action in the present context, such as CB or α2 receptor agonism. This could also explain how certain CBG dosing regimens counteract CBD and its 5-HT1A activating effects when given in combination, leading to subadditivity. In contrast, under the Day 6 dosing regimen where only CBD showed efficacy, the combination produced synergist effects associated with a 6-fold increase in potency (Figure 2). This result suggesting complementary mechanisms of action under this dosing regimen of CBD and CBG following the one acute injection, such as CBD producing 5-HT1A receptor activation and CBG producing CB and α2 receptor agonism.

We also replicated two recent reports demonstrating that CBD can attenuate opioid withdrawal signs in rodents [27,28]. Navarrete et al. [27] reported that anxiety-like behavior, motor activity, and withdrawal-related somatic signs associated with spontaneous heroin withdrawal were significantly reduced in mice treated with CBD. Moreover, CBD treatment normalized decreased gene expression of the CB1 cannabinoid receptor and increased already enhanced gene expression of the CB2 receptor in the nucleus accumbens following spontaneous heroin withdrawal. Scicluna et al. [28] demonstrated that CBD attenuated gastrointestinal symptoms associated with both precipitated and spontaneous withdrawal from oxycodone in male and female mice. Mechanistically, there is a wealth of evidence for opioid-cannabinoid crosstalk [32] and, therefore, a potential for CBD to directly interact with opioid receptors [33] or indirectly modulate the opioid system through modulation of CB1 and CB2 receptors [34] (although there remains debate regarding the direct or indirect interaction of CBD with both cannabinoid receptors, see [35,36,37]). We extended the behavioral findings to determine that pretreatment with CBD or CBG, 2 h prior to naloxone injection, significantly attenuated jumping behavior, a classical withdrawal sign observed in mice in this model. This is the first report we are aware of to investigate the effect of CBG on opioid withdrawal. Pharmacologically, α2-adrenergic agonists have been used historically to attenuate opioid withdrawal [38], so this supports our observation here that CBG was effective in this model. Similar to what was observed in the Scicluna study with CBD, neither drug was very potent or efficacious in our model. We tested up to 100 mg/kg for each compound, which was as high as we could test before encountering confounding motor and sedating effects for these drugs. Only the 100 mg/kg dose of each drug was statistically significant, and ED_25_ values instead of ED_50_ values were calculated due to limited efficacy at the dose range tested. However, the two compounds were relatively equipotent and equieffective to one another. When tested in combination based on their equivalent potency and efficacy, the results showed a robust synergy, with a greater than 4-fold increase in potency of the combination compared to its predicted interactive effect. Again, synergy suggests complementary mechanisms of action, perhaps opioid and cannabinoid-modulating effects of CBD and α2-adrenergic and cannabinoid modulating effects of CBG.

Lastly, we determined whether CBG would modify the modest antinociceptive effects of CBD or the robust antinociceptive effects of morphine. We found that CBG alone produced minimal antinociceptive effects on the hotplate, like observations with the tail flick assay reported by Sepulveda et al. (2022) [15]. CBD produced modest to moderate effects on the hotplate, as we have reported previously [12,26]. In the literature, CBD has been reported to increase [26,39], attenuate [26], or produce additive acute antinociceptive effects [15] when administration in combination with morphine, depending upon the assay and other variables. With CBG, we found a slight but statistically significant attenuation of morphine’s antinociceptive effect on the hotplate, and a slight but non-significant attenuation of CBD’s effect.

In summary, CBD and CBG share overlapping but distinct pharmacological profiles that continue to be defined. These profiles appear to be complementary to their potential therapeutic application in the context of pain and substance use disorders, as is supported by several of the combination studies presented here. However, our results also reveal dosing regimens and assays wherein the combination produces potent sub-additive effects, suggestive of pharmacological effects and mechanisms that hinder each other’s efficacy. Limitations of the current study include the exclusive use of male mice, although we have stated that we have recently published another study specifically focused on CBG and sex differences [29]. In addition, although we tried to be inclusive of dose ranges and behavioral tests used, many more dosing regimens and combination ratios can be explored, increasing translational relevance. Interestingly, in a recent published survey of patients using CBG-rich Cannabis preparations, most respondents reported greater efficacy of CBG-predominant Cannabis over conventional pharmacotherapy for the treatment of pain, anxiety, and depression [40]. Taken, together, continued characterization of the single and combined effects of CBD, CBG, and additional phytocannabinoids is necessary to determine safe and optimal phytocannabinoids approaches to treating pain, substance use disorder, and potentially many other indications.

## Figures and Tables

**Figure 1 biomedicines-12-01145-f001:**
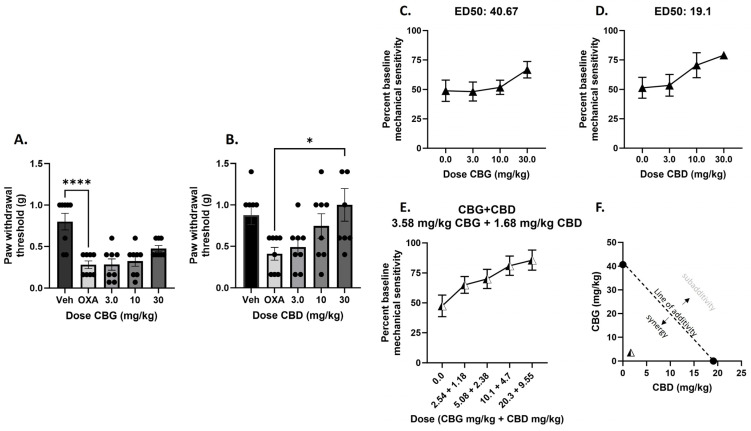
Administration of CBD, but not CBG, on Day 6 reverse mechanical sensitivity following oxaliplatin exposure in male C57BL/6 mice, and the combination of CBG and CBD administered on Day 6 produces synergistic attenuation of mechanical sensitivity associated with oxaliplatin administration in male C57BL/6 mice. (**A**,**B**) Mice were treated with vehicle or oxaliplatin (OXA) on Day 1 and either vehicle or CBG (**Left**: 3.0–30 mg/kg i.p.), or CBD (**Right**: 3.0–30 mg/kg i.p.) on Day 6. Mechanical sensitivity was measured at baseline and on Day 6, 2 h post vehicle or cannabinoid administration. Results demonstrate that only 30 mg/kg CBD administration on Day 6 significantly reverses oxaliplatin-associated mechanical sensitivity. One-way ANOVAs with Dunnett’s multiple comparison tests, * *p* < 0.05, **** *p* < 0.0001. (**C**,**D**) To determine equipotent dose ratios of CBG and CBD to test in combination and to predict additive effect level of the combination, ED_50_s were calculated for CBG and CBD. To test the combination, mice were treated with vehicle or oxaliplatin on Day 1 and increasing dose combinations of CBG and CBD in a 1:1 ratio based on potency on Day 6 (**E**). Mechanical sensitivity was measured at baseline and on Day 6, 2 h post vehicle or cannabinoid administration. Plotting the actual combined ED_50_ with the predicted combined ED_50_, the isobologram shows that the combination produces a synergistic attenuation of mechanical sensitivity, defined as below and to the left of the line of additivity (**F**). n = 8/group.

**Figure 2 biomedicines-12-01145-f002:**
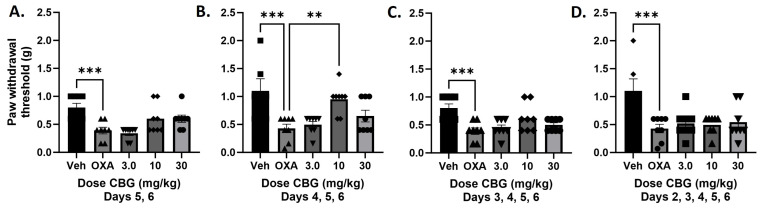
Administration of CBG for three consecutive days, but not fewer or more, produced a significant attenuation of mechanical sensitivity following oxaliplatin exposure in male C57BL/6 mice. Separate groups of mice were treated with vehicle or oxaliplatin (OXA) on Day 1 and either vehicle or CBG on Days 5 and 6 (**A**), Days 4, 5, and 6 (**B**), Days 3, 4, 5, and 6 (**C**), or Days 2, 3, 4, 5, and 6 (**D**). Mechanical sensitivity was measured at baseline and on Day 6, 2 h post vehicle or CBG administration significantly reverses oxaliplatin-associated mechanical sensitivity. One-way ANOVAs with Dunnett’s multiple comparison tests, ** *p* < 0.01, *** *p* < 0.001 (n = 8/group).

**Figure 3 biomedicines-12-01145-f003:**
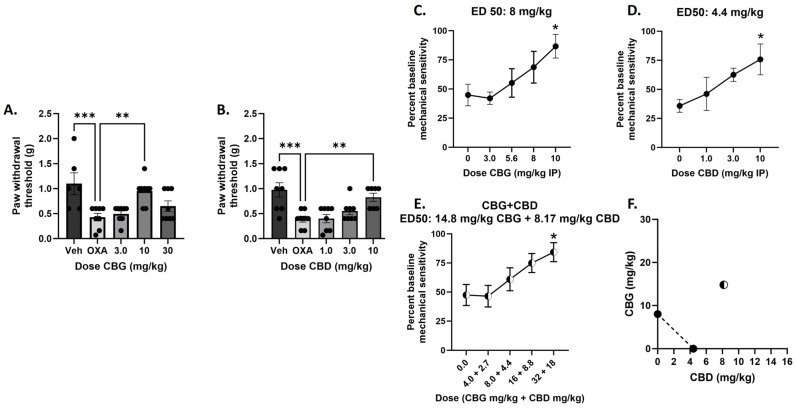
Administration of CBG or CBD on Day 4, 5, and 6 reverse mechanical sensitivity following oxaliplatin exposure in male C57BL/6 mice and a combination of CBG and CBD administered on Day 4, 5, and 6 produces subadditive attenuation of mechanical sensitivity associated with oxaliplatin administration in male C57BL/6 mice. (**A**,**B**) Mice were treated with vehicle or oxaliplatin (OXA) on Day 1 and either vehicle or CBG (**Left**: 3.0–30 mg/kg i.p.), or CBD (**Right**: 1.0–10 mg/kg i.p.) on Day 4, 5, and 6. Mechanical sensitivity was measured at baseline and on Day 6, 2 h post vehicle or cannabinoid administration. Results demonstrate that only 10 mg/kg CBG or 30 mg/kg CBD administration on Day 4, 5, and 6 significantly reverses oxaliplatin-associated mechanical sensitivity. One-way ANOVAs with Dunnett’s multiple comparison tests, ** *p* < 0.01, *** *p* < 0.001. (**C**,**D**) To determine equipotent dose ratios of CBG and CBD to test in combination and to predict additive effect level of the combination, ED_50_s were calculated for CBG and CBD. To test the combination, mice were treated with vehicle or oxaliplatin on Day 1 and increasing dose combinations of CBG and CBD in a 1:1 ratio based on potency on Day 4, 5, and 6. (**E**). Mechanical sensitivity was measured at baseline and on Day 6, 2 h post vehicle or cannabinoid administration. * *p* < 0.05. Plotting the actual combined ED_50_ with the predicted combined ED_50_, the isobologram shows that the combination produces a synergistic attenuation of mechanical sensitivity, defined as below and to the left of the line of additivity (**F**). n = 8/group.

**Figure 4 biomedicines-12-01145-f004:**
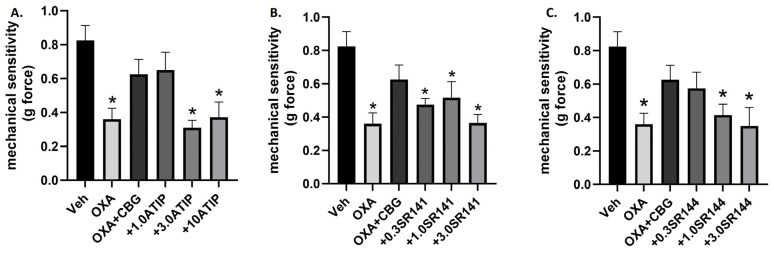
Co-administration of selective α2-adrenergic receptor antagonist atipamezole (**A**), CB1 receptor selective antagonist SR141716 (**B**), or CB2 receptor selective antagonist SR144528 (**C**) significantly attenuates the effect of CBG on mechanical sensitivity following oxaliplatin exposure in male C57BL/6 mice. Separate groups of mice were treated with vehicle or oxaliplatin (OXA) on Day 1 and either vehicle or antagonist and vehicle or CBG on Days 4, 5, and 6. Mechanical sensitivity was measured at baseline on Day 6, 2 h post vehicle or drug administration. Results demonstrate that all three antagonists significantly attenuated CBG reversal of oxaliplatin-associated mechanical sensitivity. One-way ANOVAs with Dunnett’s multiple comparison tests, * *p* < 0.05 compared with vehicle control (n = 8/group).

**Figure 5 biomedicines-12-01145-f005:**
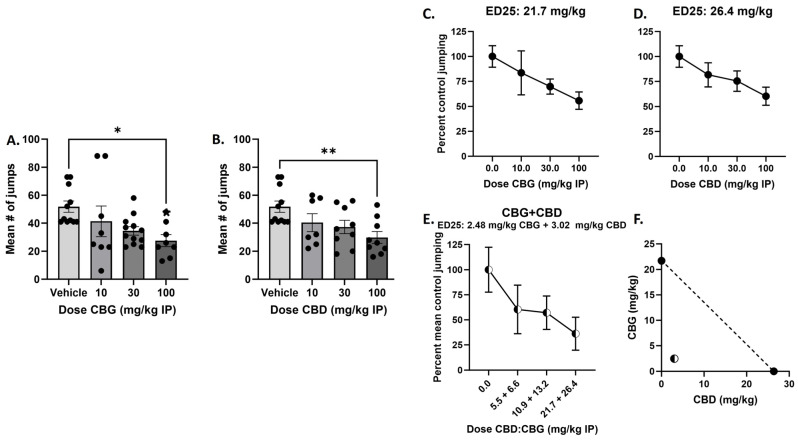
Acute administration of CBG or CBD produced a significant attenuation of naloxone-precipitated jumping behavior in male C57BL/6 mice chronically treatment with morphine, and a combination of CBG and CBD produced synergistic attenuation of naloxone-precipitated jumping behavior. (**A**,**B**) Mice were treated with increasing doses of morphine (20–100 mg/kg s.c.) twice daily for 5 days. On the morning of the last morphine injection, mice were co-administered vehicle or CBG (**Left**: 10–100 mg/kg i.p.), or CBD (**Right**: 10–100 mg/kg i.p.). Two hours later, all mice were injected with naloxone (3.0 mg/kg s.c.) and jumping behaviors was immediately observed for 30 min. Results demonstrate that 100 mg/kg CBG or CBD administration significantly attenuated naloxone-precipitated jumping. One-way ANOVAs with Dunnett’s multiple comparison tests, * *p* < 0.05, ** *p* < 0.01 (n = 7–10/group). (**C**,**D**) To determine equipotent dose ratios of CBG and CBD to test in combination and to predict additive effect level of the combination, ED_25_s were calculated for CBG and CBD. (**E**) To test the combination, mice were treated with increasing dose combinations of CBG and CBD in a 1:1 ratio based on potency, concomitant with the final morphine injection. Two hours later, all mice were injected with naloxone (3.0 mg/kg s.c.) and jumping behavior was immediately observed for 30 min. Plotting the actual combined ED_25_s with the predicted combined ED_25_s, the isobologram shows that the combination produces a synergistic attenuation of jumping behavior (**F**). (n = 8/group).

**Figure 6 biomedicines-12-01145-f006:**
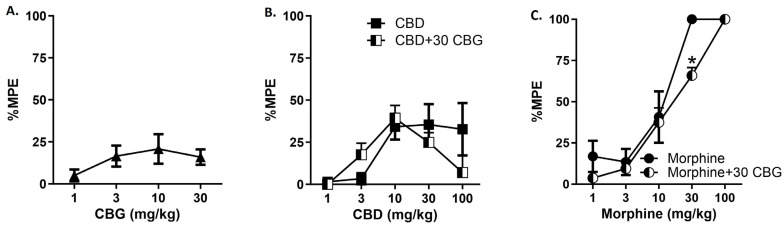
CBG attenuates thermal antinociceptive effects of morphine or CBD in male C57BL/6 mice. The effect of ascending doses of CBG (**A**), CBD (**B**), or morphine (**C**) to produce thermal antinociception on the 56 °C hotplate was determined. Results showed approximate 20% maximum possible effect for CBG, 35% MPE for CBD, and 100% MPE for morphine. A fixed dose of 30 mg/kg CBG was selected to test in combination with the CBD (**B**) and morphine (**C**) dose response curves and results demonstrate that CBG attenuated the antinociceptive effects of morphine, with a statistically significant effect at 30 mg/kg. Two-way ANOVAs with Sidak’s multiple comparison tests, * *p* < 0.05 (n = 8/group).

## Data Availability

Datasets available by request.

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
