# Peer review of "Single and Combined Effects of Cannabigerol (CBG) and Cannabidiol (CBD) in Mouse Models of Oxaliplatin-Associated Mechanical Sensitivity, Opioid Antinociception, and Naloxone-Precipitated Opioid Withdrawal"

_biomedicines, 2024, doi:10.3390/biomedicines12061145_

Round 1
Reviewer 1 Report
Comments and Suggestions for Authors
In this manuscript, Hayduk et al. report that both cannabigerol (CBG) and cannabidiol (CBD) alleviate oxaliplatin-associated mechanical sensitivity with different doses and regimens required for each compound. Von Frey was used as the primary readout for these measures. Additionally, they demonstrate that the two work synergistically for acute treatment, but sub-additively for sub-chronic (3-day) treatment. They then show that CBG and CBD decrease naloxone-precipitated withdrawal jumping behavior alone (at relatively high doses) and synergistically together. Lastly, they report that CBG attenuates the acute antinociceptive effects of morphine and CBD, although this appears to be mild.
The paper is well-written, and the conclusions are supported by the data. The topic is timely and important in the context of alternatives to opioids. Although the results are somewhat contrasting to one another, (synergy in one context, subadditive in the next), the authors do a good job of explaining this in the discussion. Slightly more discussion could be devoted to the potential for human use (since flower/mixtures are usually administered chronically for these pain conditions), and some figure consolidation/consistency may be warranted. Issues are noted below (all relatively minor), which should be addressed prior to publication.
1) If these mice were purchased from Taconic, I believe the correct naming is C57BL/6NTac. The J designation is for Jax.
2) A large focus of both the intro and conclusions is on receptor-based mechanism of action of CBG. Especially for the oxaloplatin experiments, it could be useful to include one experiment demonstrating that a2/CB1/CB2 antagonists block this effect. This would help bridge the language in these parts of the paper with the results, although it’s not a requirement for publication.
3) Figure 1- The Oxali data on the right (CBD) does not report as significant compared to vehicle. Is this true? The “OXALI” term could be shortened (OXA) so it does not get visually confused with “veh”.
4) Figure 2- "Dose" should be capitalized to be consistent. The circle on the additive line is confusing, because it looks like data. Is it data? Further, the word synergy is directly next to the actual data (I think), which makes this whole plot somewhat hard to interpret. I would move the words so they are not directly adjacent to the data, and perhaps remove the white/black circle. Another way of making synergy clear is to draw an arrow from the additive line to the square, or from the circle to the square?
5) Generally, I think figures 1 and 2 could be combined, as well as 4 and 5, and 6 and 7. These are essentially the same data, just plotted as a percent of control.
Also, there is general inconsistency in the color scheme used for combination experiment plotting. Some are triangles, some are circles, some are mixed color, some are just black for combination. I suggest using circles in black (CBG), and white (CBD), with the combination as half-black, half-white (like figure 7C). The phrase “IP” could also be removed from every x-axis where it is present.
6) Figure 8- Several different pieces of terminology are used here that require clarification. The results texts says post-hoc tests were Sidak’s but the figure legend says Tukey’s. The star in the figure, which I assume from the results text is the difference at 30 mg/kg morphine vs morphine cbg, appears to be in the wrong place. Finally, the legend text talks of a significant shift of the dose response curve to the right. I don’t believe you can conclude that from this analysis. It also states that “CBG attenuated the antinociceptive effect their highest doses” which is not true since at the highest dose there is no difference.
Another way of displaying this graph is to include CBG in panels B and C, and simply omit D. This would make the data more visually clear as they are separated. Especially as there is not a statistical reason to have all 4 curves on one graph.
7) With respect to the discussion, I think slightly expanding why these dose combinations might be beneficial or not in certain cancer-treatment settings in humans is important for the context of the paper. Also, there is no discussion of how CBG may be attenuating morphine’s effect at a receptor/mechanistic level.
Author Response
Thank you so much for the very helpful review!
1) If these mice were purchased from Taconic, I believe the correct naming is C57BL/6NTac. The J designation is for Jax.
We have changed this.
2) A large focus of both the intro and conclusions is on receptor-based mechanism of action of CBG. Especially for the oxaloplatin experiments, it could be useful to include one experiment demonstrating that a2/CB1/CB2 antagonists block this effect. This would help bridge the language in these parts of the paper with the results, although it’s not a requirement for publication.
We have added this.
3) Figure 1- The Oxali data on the right (CBD) does not report as significant compared to vehicle. Is this true? The “OXALI” term could be shortened (OXA) so it does not get visually confused with “veh”.
We changed this.
4) Figure 2- "Dose" should be capitalized to be consistent. The circle on the additive line is confusing, because it looks like data. Is it data? Further, the word synergy is directly next to the actual data (I think), which makes this whole plot somewhat hard to interpret. I would move the words so they are not directly adjacent to the data, and perhaps remove the white/black circle. Another way of making synergy clear is to draw an arrow from the additive line to the square, or from the circle to the square?
Fixed.
5) Generally, I think figures 1 and 2 could be combined, as well as 4 and 5, and 6 and 7. These are essentially the same data, just plotted as a percent of control.
We have combined these.
Also, there is general inconsistency in the color scheme used for combination experiment plotting. Some are triangles, some are circles, some are mixed color, some are just black for combination. I suggest using circles in black (CBG), and white (CBD), with the combination as half-black, half-white (like figure 7C). The phrase “IP” could also be removed from every x-axis where it is present.
Fixed, thanks!
6) Figure 8- Several different pieces of terminology are used here that require clarification. The results texts says post-hoc tests were Sidak’s but the figure legend says Tukey’s. The star in the figure, which I assume from the results text is the difference at 30 mg/kg morphine vs morphine cbg, appears to be in the wrong place. Finally, the legend text talks of a significant shift of the dose response curve to the right. I don’t believe you can conclude that from this analysis. It also states that “CBG attenuated the antinociceptive effect their highest doses” which is not true since at the highest dose there is no difference.
Fixed, thanks!
Another way of displaying this graph is to include CBG in panels B and C, and simply omit D. This would make the data more visually clear as they are separated. Especially as there is not a statistical reason to have all 4 curves on one graph.
Fixed
7) With respect to the discussion, I think slightly expanding why these dose combinations might be beneficial or not in certain cancer-treatment settings in humans is important for the context of the paper. Also, there is no discussion of how CBG may be attenuating morphine’s effect at a receptor/mechanistic level.
We added comment on CBG rich cannabis efficacy in patients, and had already addressed CBG alpha adrengergic effects for opioid withdrawal fitting with current practices.
Reviewer 2 Report
Comments and Suggestions for Authors
Cannabis has been used for intoxication but also for medicinal purposes by various human groups since prehistoric times. The modern use of hemp in medicine is not an easy issue. This is due to many problems presented by modern medicine. These include the validity of clinical trials, unmet medical needs, but above all, communication between the medical community and society. Of all the active ingredients found in Cannabis sp., phytocannabinoids are the most interesting. Over 70 different phytocannabinoids are already known, but the most important compounds belonging to this group are delta-9-tetrahydrocannabinol (Δ-9-THC), cannabidiol (CBD), cannabidiolic acid (CBDA), cannabinol (CBN) and cannabigerol (CBG). CBG is a naturally occurring cannabinoid in cannabis. It is a precursor to other cannabinoids, including CBD and THC, which means its presence is crucial for the synthesis of other compounds in the plant.
Chemotherapy-induced peripheral neurotoxicity (CIPN) is one of the most frequent side effects of antineoplastic treatment, particularly of lung, breast, prostate, gastrointestinal, and germinal cancers, as well as of different forms of leukemia, lymphoma, and multiple myeloma. Currently, no effective therapies are available for CIPN prevention, and symptomatic treatment is frequently ineffective; thus, several clinical trials are addressing this unmet clinical need. Among possible pharmacological treatments of CIPN, modulation of the cannabinoids might be particularly promising, especially in those CIPN types where analgesia and neuroinflammation modulation might be beneficial.
Therefore, work assessing the single and combined effects of cannabigreol and cannabidiol is a very important source of information. In my opinion current manuscript concerns an extremely important and contemporary issue. The abstract is a summary of all elements of the manuscript. The introduction presents all necessary information and ends with a well-formulated goal of the experiments. Numerous figures presenting the experimental results additionally increase the value of the manuscript. The discussion explains research conducted by the Authors.
However, to improve the quality and value of the manuscript in my opinion:
In the materials and methods section, the Authors should include information on the basis of which research the doses of used substances were selected.
It is also necessary to describe the groups that were used in the experiment.
The Authors should add limitations of the study and conclusions as a separated point.
Additionally, in each experiment it is necessary to determine how many animals were eliminated from the calculations. This is very important, especially since the groups contained 8 animals, while the values given for the ANOVA analyzes include 35 or 35 and not 40. Why were individual individuals eliminated and from which groups?
The entire manuscript concerns an important and interesting problem, but its correct understanding is significantly hampered by the chaos in describing the content. Results should be described in individual, clearly demarcated and named points.
Author Response
In the materials and methods section, the Authors should include information on the basis of which research the doses of used substances were selected.
-- We have added this, thanks
It is also necessary to describe the groups that were used in the experiment.
--I do not understand this comment. I believe the groups are very clear in the methods, results and graphs
The Authors should add limitations of the study and conclusions as a separated point.
--this has been added
Additionally, in each experiment it is necessary to determine how many animals were eliminated from the calculations. This is very important, especially since the groups contained 8 animals, while the values given for the ANOVA analyzes include 35 or 35 and not 40. Why were individual individuals eliminated and from which groups?
--the only mice that were excluded were mice that were euthanized due to fighting wounds. No data were excluded from the analyses, we have added this